# Anaerobic Feces Processing for Fecal Microbiota Transplantation Improves Viability of Obligate Anaerobes

**DOI:** 10.3390/microorganisms11092238

**Published:** 2023-09-05

**Authors:** Mèlanie V. Bénard, Iñaki Arretxe, Koen Wortelboer, Hermie J. M. Harmsen, Mark Davids, Clara M. A. de Bruijn, Marc A. Benninga, Floor Hugenholtz, Hilde Herrema, Cyriel Y. Ponsioen

**Affiliations:** 1Department of Gastroenterology and Hepatology, Amsterdam Gastroenterology Endocrinology Metabolism (AGEM), Amsterdam UMC, University of Amsterdam, 1105 AZ Amsterdam, The Netherlands; m.v.benard@amsterdamumc.nl (M.V.B.); iarretxe98@gmail.com (I.A.); c.m.debruijn@amsterdamumc.nl (C.M.A.d.B.); m.a.benninga@amsterdamumc.nl (M.A.B.); 2Department of Endocrinology and Metabolism, Amsterdam Gastroenterology Endocrinology Metabolism (AGEM), Amsterdam UMC, University of Amsterdam, 1105 AZ Amsterdam, The Netherlands; 3Department of Experimental Vascular Medicine, Amsterdam Cardiovascular Sciences (ACS), Amsterdam Gastroenterology Endocrinology Metabolism (AGEM), Amsterdam UMC, University of Amsterdam, 1105 AZ Amsterdam, The Netherlands; k.wortelboer@amsterdamumc.nl (K.W.); m.davids@amsterdamumc.nl (M.D.); h.j.herrema@amsterdamumc.nl (H.H.); 4Department of Medical Microbiology and Infection Prevention, University of Groningen, University Medical Center Groningen, 9700 RB Groningen, The Netherlands; h.j.m.harmsen@umcg.nl; 5Pediatric Gastroenterology, Hepatology and Nutrition, Emma Children’s Hospital, Amsterdam UMC, University of Amsterdam, 1105 AZ Amsterdam, The Netherlands; 6Amsterdam Reproduction & Development Research Institute, Emma Children’s Hospital, Amsterdam UMC, University of Amsterdam, 1105 AZ Amsterdam, The Netherlands; 7Center for Experimental and Molecular Medicine, Amsterdam Medical Center, Amsterdam UMC, University of Amsterdam, 1105 AZ Amsterdam, The Netherlands; floorhugenholtz@gmail.com

**Keywords:** fecal microbiota transplantation, culturability, bacterial viability, gut microbiota, ulcerative colitis, anaerobic bacteria, sample processing, culturing of fecal microbiota, next-generation sequencing, frozen microbiota

## Abstract

Fecal microbiota transplantation (FMT) is under investigation for several indications, including ulcerative colitis (UC). The clinical success of FMT depends partly on the engraftment of viable bacteria. Because the vast majority of human gut microbiota consists of anaerobes, the currently used aerobic processing protocols of donor stool may diminish the bacterial viability of transplanted material. This study assessed the effect of four processing techniques for donor stool (i.e., anaerobic and aerobic, both direct processing and after temporary cool storage) on bacterial viability. By combining anaerobic culturing on customized media for anaerobes with 16S rRNA sequencing, we could successfully culture and identify the majority of the bacteria present in raw fecal suspensions. We show that direct anaerobic processing of donor stool is superior to aerobic processing conditions for preserving the bacterial viability of obligate anaerobes and butyrate-producing bacteria related to the clinical response to FMT in ulcerative colitis patients, including *Faecalibacterium*, *Eubacterium hallii*, and *Blautia*. The effect of oxygen exposure during stool processing decreased when the samples were stored long-term. Our results confirm the importance of sample conditioning to preserve the bacterial viability of oxygen-sensitive gut bacteria. Anaerobic processing of donor stool may lead to increased clinical success of FMT, which should further be investigated in clinical trials.

## 1. Introduction

Fecal microbiota transplantation (FMT)—the transfer of processed stool from a healthy donor into a diseased recipient—has an established role as a rescue treatment for recurrent *Clostridioides difficile* infection (rCDI), with high cure rates reaching ~90–95% [1,2]. Donor microbiome composition and variations in stool-processing protocols seem to have little impact on the clinical efficacy of FMT for this indication [2,3,4]. In addition to rCDI, FMT has been investigated for multiple indications, including the inflammatory bowel disorder ulcerative colitis (UC). The efficacy of donor FMT for inducing remission in UC has been demonstrated in several randomized controlled trials (RCTs), with variable remission rates between 24 and 53% [5,6,7,8]. However, other studies have failed to replicate this effect [9,10], and this discrepancy is possibly due to the large heterogeneity in stool processing protocols, treatment schedules, and donor selection.

Because FMT targets the gut microbiota, it is expected that the engraftment of “beneficial” and/or the replacement of “unfavorable” microbes and their metabolic functions causes the reported effects [11,12]. The metabolites of interest include short-chain fatty acids (SCFAs), in particular butyrate, which serves as a primary energy source for the intestinal epithelial cells and has widely reported anti-inflammatory properties [13,14]. The clinical success of FMT has been described to relate to higher donor strain engraftment [15], which in turn is largely dependent on the taxonomic composition of recipient and donor and the abundance of species [16].

Dysbiosis in UC patients is marked by an overall decreased bacterial diversity, a decreased abundance of strict anaerobic butyrate producers (including the Firmicutes members *F. prausnitzii* and *E. rectale*), and an increased abundance of facultative anaerobes (such as *E. coli* from the family Enterobacteriaceae) [17]. In line with these findings, the clinical response to donor FMT in UC patients has been found to relate to an increase in alpha-diversity, SCFA production, and an abundance of the obligate anaerobes Clostridiales Cluster IV/XIVa and Bacteroides post-FMT, whereas nonresponse is related to an increase in Proteobacteria and Fusobacteria [18]. Several FMT studies for UC report a donor-microbiome-dependent effect, suggesting that specific bacterial donor strains play a critical role in the treatment effect of FMT in this disease [5,6,19,20]. Conversely, a recent meta-study assessing several FMT indications did not find a correlation between higher donor strain engraftment and clinical success of FMT [21], highlighting the importance of other determinants such as donor–recipient complementarity [21] and procedural factors.

Anaerobic bacteria account for the vast majority of the human gut microbiome [22,23]; oxygen exposure can inhibit the growth of facultative anaerobes and can be lethal to obligate anaerobes [24,25]. Therefore, commonly applied aerobic stool processing methods may result in decreased levels of viable bacteria, especially of obligate anaerobes. This loss of anaerobes could potentially destroy beneficial trophic networks and lead to impaired donor–recipient complementarity and low engraftment. If anaerobic preparation facilitates the retention of anaerobes, this may improve FMT outcomes in UC and potentially also in other indications [26].

Currently, there is no consensus on the optimal stool processing method for FMT beyond rCDI; the protocols practiced in clinical trials for several indications therefore differ widely [27]. Despite the notion of the importance of obligate anaerobes for clinical FMT success (in UC), there have been only a limited number of reports on the effects of oxygen exposure during FMT preparation on the viability and composition of processed donor microbiota [28,29,30,31]. Characterization of the donor microbiota in FMT trials is usually performed on feces samples with high throughput sequencing. However, this technique is not suitable to assess the viability of bacteria because it targets DNA of both viable and nonviable organisms. There are several strategies to overcome this problem [32,33,34], one being a combination of sequencing techniques with culturing methods to grow viable bacteria.

In the present study, we compared anaerobic and aerobic preparation techniques for the production of frozen fecal microbiota suspensions. By combining two culturing experiments with next-generation sequencing, we report the effects of three potential sources of degradation on bacterial viability: oxygen exposure, lag time between stool donation and processing, and long-term storage. We hypothesized that direct anaerobic processing is superior to preserve the viability of essential obligate anaerobic strains compared with aerobic processing protocols.

## 2. Materials and Methods

### 2.1. Stool Sample Collection and Processing to Fecal Microbial Transplant

A schematic representation of the processing protocol is depicted in Figure 1. Eight fresh stool samples were collected from six human volunteers, none of whom suffered from any gastrointestinal tract disorder (Appendix A). Volunteers who took antibiotics within three months prior to donation or within the preceding three months before sampling could not participate in the study. Two volunteers donated at two time points. One participant (#4) was an active stool donor for an ongoing clinical FMT trial and was screened as described previously [35].

Stool was collected onsite in a plastic container placed in a 0.6 L vacuum box (10006101;0 from ALLPAX, Papenburg, Germany. Immediately after collection, two anaerobic gas generator sachets (AnaeroGen Compact, AN0010C from Thermo Scientific, Waltham, MA, USA) were added to the vacuum box, avoiding direct contact with the stool, after which the box was vacuumed (Vacuum machine P355, 10006616;0 from ALLPAX). Within 30 min, the box was transferred and opened inside an anaerobic chamber (Anaerobic workstation Concept 500, Baker Ruskin, Sanford, ME, USA) in a nitrogen (85%), hydrogen (10%), and carbon dioxide (5%) atmosphere at +37 °C. 

The stool sample was then split into four equal subsamples to assess four FMT processing conditions: AN_0_, direct anaerobic processing; AE_0_, direct aerobic processing; AN_2.5_, anaerobic processing after 2.5 h of anaerobic storage at +4 °C; and AE_2.5_, aerobic processing after 2.5 h of aerobic storage at +4 °C. The subsamples of the anaerobic and aerobic conditions (both 0 and 2.5 h of cool storage before processing) were simultaneously processed by two researchers in an anaerobic chamber and in a sterile hood, respectively. To assess the effects of cool storage for both the anaerobic and aerobic processing conditions, subsample AN_2.5_ was kept for 2.5 h at +4 °C in a vacuumed stool container containing two new AnaeroGen sachets and prepared inside the anaerobic chamber, and subsample AE_2.5_ was exposed for 2.5 h to atmospheric oxygen at +4 °C.

The subsamples were diluted with sterile saline in a 1:6.2 feces weight (g) to saline (mL) ratio to be suitable for filtration and homogenized for 2 min using a sterile blender. Gauze filtration was performed to remove large aggregates, whereafter 85% glycerol was added to the saline stool filtrate in a 1:8.5 glycerol (mL) to filtrate (mL) ratio in order to reach a final glycerol concentration of approximately 10%, as suggested by European consensus reports [36,37]. After filtration, the fecal suspension was divided into equal aliquots (in 60 cc sterile Nutrifit/ENFit syringes with cap fully wrapped with aluminum foil to protect the content from UV light) and then frozen and stored at −80 °C. All storage material and the saline and glycerol were placed in the anaerobic chamber at least 48 h before use.

### 2.2. Quantification of Live Bacteria with CFU Counts in Short-Term- and Long-Term-Stored Samples Using Routine Nonselective Media

Culturing of the fecal microbiota transplant samples from four processing conditions from eight human stool samples (two-year storage for stool samples #1–#4 and one-month storage for stool samples #5–#8) was performed in the Amsterdam UMC, location AMC. For this culturing experiment, Columbia sheep blood agar plates (Biomérieux) were utilized. Columbia sheep blood agar plates are nonselective media, enriched for a wide variety of microorganisms. Briefly, samples were serially diluted from 10^−1^ to 10^−8^ in phosphate-buffered saline (PBS) (Fresenius Kabi, Bad Homburg, Germany). After homogenization, 20 µL of each homogenized dilution was spread on Columbia sheep blood agar plates, as shown in Figure 1. This process was carried out in duplicate. Three culturing conditions were compared: full aerobic conditions, i.e., both the preparation of samples and incubation in aerobic conditions (flow hood and incubator); partial anaerobic conditions, i.e., short preparation in atmospheric oxygen and incubation under anaerobic conditions using AnaeroPack^®^ (Tokyo, Japan); and full anaerobic conditions, i.e., preparation and incubation under anaerobic conditions (using anaerobic chamber Concept 500 and AnaeroPack^®^). The oxygen concentration in the anaerobic chamber during the dilution preparation and plating was between 0.0 and 0.3% O_2_. Colony-forming units (CFUs) were counted after 24 h for the aerobic culturing conditions and after 48 h for the anaerobic culturing conditions.

### 2.3. Microbial Culturomics Using Customized Media for Anaerobes

Culturing of fecal microbiota transplant samples #1 to #4 from the four processing conditions was performed in the UMCG (University Medical Center Groningen). Anaerobic plating and incubation were performed in an anaerobic chamber (Whitley A35, Don Whitley Scientific) in a nitrogen (80%), hydrogen (10%), and carbon dioxide (10%) atmosphere at +37 °C. Samples were thawed and plated on two different agar media. Both media consisted of YCFA (yeast, casitone, and fatty acids prepared as described previously [38]) with different selective carbohydrates added: one medium contained 4 g/L glucose-cellobiose-Hi-maize in a 1:1 ratio (2 g each) (GC medium) and the other contained 4 g/L apple and kiwi pectin in a 1:1 ratio (P medium). YCFA is an enriched nonselective medium suitable for cultivation of most anaerobic bacteria. On both the GC medium and the P medium, 200 μL of thawed fecal sample was plated. All plating was performed in duplicate. After 48 h of incubation, a full inoculation loop of the growth with a representative was smeared onto new plates. The growth on these plates (after 48 h), as a representation of viable bacteria, was used for 16S rRNA gene sequencing. Raw samples of the feces suspensions were also analyzed with DNA sequencing and are further referred to as “uncultured samples”.

### 2.4. DNA Extraction, PCR, and 16S rRNA Gene Sequencing

**Routine media:** For all processing conditions from stool samples #5–#8, a sample of the lowest dilution (10–1) was taken before and after incubation. Deoxyribonucleic acid (DNA) was extracted from 0.25 g fecal sample or the harvest of the full-grown plate in physiological salt, using a QIAamp DNA Stool Mini Kit (Qiagen, Hilden, Germany). Modified barcoded 341F and 806R primers were used to amplify the V3–V4 region of the 16S rRNA gene. Details on the PCR, barcoded primers, and sequencing library preparation were described previously [39]. The amplicons were sequenced with an MiSeq Illumina sequencing platform. The paired-end reads, demultiplexed based on barcode, were retrieved from the Illumina platform. Reads were trimmed, filtered by quality, and assigned to OTU with the CLC genomic workbench (Qiagen) using the SILVA [40] reference database (version 132) for taxonomy.

**Customized media:** DNA was extracted using a repeated bead-beating protocol. The extracted DNA was isolated and purified by using Maxwell^®^ RSC Blood DNA kits (Promega, Madison, WI, USA). DNA concentration was measured with the Qubit^®^ dsDNA BR kit (ThermoFisher, Waltham, MA, USA). The isolated DNA was stored at −80 °C. Using a single-step polymerase chain reaction (PCR) protocol targeting the V3–V4 region, 16S rRNA gene amplicons were generated. Sequencing was performed on an Illumina MiSeq platform. Taxonomy was assigned using the SILVA reference database (version 138). Obtained sequence reads were processed using a VSEARCH [41] (version 2.15.2)-based pipeline. Paired end reads were merged, with max differences set to 100 and allowing for staggered overlap. Amplicon sequence variants (ASVs) were inferred from reads with a lower than 1.5 expected error rate using the cluster_unoise with centroids algorithm with a minsize of 4, after which chimeras were removed using the UCHIME3 de novo method. For each sample, ASV abundances were determined by mapping the merged reads against the ASV sequences with identical matches sequence set using the usearch_global algorithm, with a 0.97 distance cutoff. Taxonomy was assigned using R (version 4.0.5) and the dada2 [42] assign-taxonomy function using the SILVA (version 132) reference database.

### 2.5. Statistical Analysis of Community Composition and CFU Counts

The obtained count tables were rarefied to 30,000 counts. Multi-level PCA was used to visualize intra-donor differences in culturing. Differences in the viability of the taxa were determined using linear mixed effect models. The alpha-diversity (Shannon index) and beta-diversity (Euclidean distance) were computed using R (V4.0.5). In both experiments, bacterial taxa were allocated at the genus level as “anaerobic”, “aerotolerant”, or “facultative”, and as “butyrate producer” or “non-butyrate-producer”, by an expert microbiologist (HJMH) and bio-informatician (MD) (Appendix A).

The CFU data were not normally distributed and therefore were analyzed with nonparametric tests. Data were log transformed. Single comparisons between two conditions were performed with Wilcoxon matched-pairs tests. Comparisons between the anaerobic and aerobic processing conditions (both direct processing and temporarily stored combined) were also performed with the Wilcoxon matched-pairs test. A *p*-value < 0.05 was considered statistically significant. A general linear mixed model was used to evaluate fixed effects of intercept, atmosphere, (in)direct processing, and storage time on the log CFU counts, with a random effect of the volunteer. Statistical analysis of the CFU counts was performed with IBM SPSS Statistics (version 28.0.1.1). Graphs for CFU data were obtained with GraphPad Prism (version 9.1.0.).

## 3. Results

### 3.1. Quantification of Bacterial Viability Using CFU Counts

We compared the bacterial viability of eight human stool samples after either anaerobic or aerobic processing of frozen fecal microbiota suspensions, either processed directly after donation or temporarily stored before processing (Figure 1). Strict anaerobic culturing with routine sheep blood agar plates resulted in higher CFU counts in the anaerobic processing conditions (median of 10.31 log CFU/g feces, IQR: 0.18) compared with the aerobic conditions (median of 10.16 log CFU/g feces, IQR: 0.37, *p* < 0.001). Comparing the four processing conditions, direct anaerobic processing led to significantly higher CFU counts compared with both direct aerobic processing and aerobic processing after temporary storage (Figure 2, *p* < 0.05). No significant effect was found between the direct and indirect processing conditions (after 2.5 h of storage) for either the anaerobic (AN_0_ vs. AN_2.5_) or aerobic storage conditions (AE_0_ vs. AE_2.5_).

Linear mixed-model analysis revealed that atmosphere (i.e., anaerobic or aerobic) had a significant main effect on the CFU counts (Appendix A, *p* < 0.001). This effect was modified by storage time, where the differences between the anaerobic and aerobic processing conditions were larger for the one-month-stored samples compared with the two-years-stored samples (interaction effect, *p* < 0.01).

Sub-analysis of the samples based on storage time (N = 4 for both groups) showed that for the one-month-stored samples only, significantly higher CFU counts were found in the anaerobic processing conditions compared with the aerobic conditions, whereas differences in the two-years-stored samples did not reach statistical significance (Appendix A).

The eight stool samples were produced by six donors; to check for a potential donor-dependent effect, a sensitivity analysis was performed with the average values of dual samples produced by two individual donors (with nearly two years in between production). This analysis showed similar results, with the highest CFU counts after direct anaerobic processing (Appendix A). As expected, aerobic culturing resulted in significantly lower CFU counts than anaerobic culturing across the four FMT preparation conditions (Appendix A). The specific CFU counts per volunteer, stool processing condition, and culturing condition are depicted in Appendix A.

### 3.2. Culturing Combined with 16S rRNA Sequencing

#### 3.2.1. Alpha- and Beta-Diversity Metrics

Anaerobic culturing was performed with routine blood agar media (Figure 3) or customized media for anaerobes (Figure 4) and combined with 16S rRNA sequencing to identify the live bacterial composition. For each medium used, the α-diversity (Shannon index) of viable taxa in the specimens processed under anaerobic conditions was significantly higher than in the specimens processed under aerobic conditions (Figure 3a and Figure 4a). Next, multi-level PCA was performed to show differences in β-diversity, while the differences were tested using PERMANOVA. This analysis revealed a clear influence of oxygen exposure, reflected in the separation on the first principal component between samples that were anaerobically processed compared with samples that were aerobically processed (Figure 3b and Figure 4b; *p* < 0.001 and *p* < 0.001). In contrast, lag time up to 2.5 h before processing did not have a significant impact on the β-diversity (Figure 3b and Figure 4b; *p* = 0.983 and *p* = 0.974).

#### 3.2.2. Butyrate-Producing Bacteria and Obligate Anaerobes

The relative abundance of the bacteria of interest (i.e., butyrate producers and obligate anaerobes) was significantly higher after anaerobic processing than aerobic processing, as assessed by culturing on blood agar media (Figure 3c,d). This effect was found with both anaerobic culturing techniques, showing that short oxygen exposure during preparation for culturing (as applied with partial anaerobic culturing) did not have a significant impact on the viability of these microbes. The customized P media and GC media for the growth of anaerobes resulted in overall higher relative abundances of butyrate producers compared with the blood agar media, with significant differences between anaerobic and aerobic processing for the GC medium only (Figure 4c).

### 3.3. Viable Bacterial Composition

Differential viable bacterial abundance associated with processing condition was assessed using linear mixed models and visualized with volcano plots (Figure 3e and Figure 4e). Processing condition had a significant effect on the relative viability of the majority of the identified genera in both culturing experiments (Appendix A). Anaerobic processing had the strongest positive effect on the viability of bacteria belonging to the oxygen-sensitive families Lachnospiraceae and Ruminococcaceae within the order Clostridiales (routine and customized media). On the genus level, anaerobic processing preserved several taxa of interest, including *Faecalibacterium* (routine media), *Eubacterium hallii* (reclassified as *Anaerobutyricum hallii* [43], customized media), and *Blautia* (routine and customized media). Conversely, aerobic processing was associated with statistically significantly increased growth of certain bacteria that have been related to nonresponse to FMT in UC [18], including *Megamonas*, *Streptococcus* (routine media), *Haemophilus* and *Ruminococcus gnavus* (customized media), and *Veillonella*, *Escherichia/Shigella*, and *Staphylococcus* (routine and customized media).

## 4. Discussion

Despite the notion that the absolute majority of gut microbiota are anaerobes, stool processing for FMT is nowadays commonly performed in the presence of atmospheric oxygen. Combining different culture techniques with 16SrRNA sequencing, our results clearly show that atmospheric oxygen exposure during stool processing significantly diminishes and alters the viable bacterial microbial community. The applied anaerobic stool processing protocol in the current study conserved the viability of obligate anaerobes—such as the Clostridia class and the family *Lachnospiraceae* from the phylum Firmicutes—that have previously been related to clinical response in the treatment of UC [18]. We found that long-term storage of frozen suspensions also had a diminishing effect on bacterial viability, whereas short, cool anaerobic storage before stool processing did not show a substantial influence on the viability of the fecal microbiota community. Our results indicate that current aerobic FMT preparation practices may compromise the therapeutic efficacy of FMT in indications other than rCDI, where donor microbiome composition and variations in stool processing protocols have little impact on the clinical efficacy of FMT [2,3,4].

Limited reports are available that evaluate the effect of different FMT preparation protocols on bacterial viability of frozen material [28,31], and there are no studies to date assessing the influence of oxygen exposure on the culturability and viability of long-term frozen FMT products. Frozen FMT is the current standard for treating rCDI and is also commonly applied for other indications under investigation in clinical trials, as it has similar efficacy rates as fresh FMT [1,3] but offers important practical and safety advantages [37]. Our study demonstrates that FMT material stored for two years still contained culturable organisms but no longer showed a significant difference between samples prepared with and without the presence of atmospheric oxygen. The difference in the effect on viability between short-term and long-term storage might be explained by gradual oxygen exposure during long-term storage as a result of diffusion through the plastic storage material [44,45], leading to the loss of obligate anaerobes after long-term storage [46]. This finding further implies that after anaerobic processing, FMT material should be used after short storage times, to take potential advantage of the higher viability of anaerobes.

We observed that anaerobic processing conserved higher abundances of several bacterial taxa of interest, including the obligate anaerobes *Faecalibacterium,* from the Oscillospiraceae (formerly known as Clostridium cluster IV), and *Dorea* and *Blautia*, belonging to the Lachnospiraceae (formerly known as Clostridium cluster XIVa). These Clostridia clades contain major groups of obligate anaerobes that produce SCFAs, including butyrate, and we have shown previously that these are associated with sustained clinical effect after FMT in UC [47]. Our sequencing results are in line with other reports that likewise noted that anaerobic stool processing is related to the retention of *Clostridium* cluster IV [28] and XIVa [28,29] and, more specifically, the anti-inflammatory bacterial genus *Faecalibacterium* [31]. A paper by Papanicolas et al. further confirmed higher butyrate production in anaerobically processed FMT compared with aerobically processed FMT by assessing the butyryl-CoA:acetate CoA-transferase gene and the direct butyrate levels in fresh FMT samples [28]. Our anaerobic processing protocol retained the viability of potentially beneficial taxa in the context of treating UC. In contrast, aerobic processing was related to certain undesirable bacterial genera that have been related to treatment nonresponse in UC patients [18], such as *Dialister, Veillonella, Escherichia* [48], and *Streptococcus* [48,49]. Because anaerobic bacteria account for the vast majority of the gut microbiota, and clinical success is partly related to the engraftment and outgrowth of viable bacteria, the advantages of anaerobic processing may extend beyond the indication of UC.

We found that anaerobic cool storage for 2.5 h before further processing as applied in our study yielded similar results compared to direct anaerobic processing. In agreement with our study, Chu et al. [31] applied DNA sequencing with propidium monoazide (PMA) to target viable bacteria in frozen fecal material and stated that the lag time before processing (up to seven hours) had a much smaller effect on bacterial viability and composition than oxygen exposure [31]. This finding has important practical implications for FMT preparation and logistic procedures for stool donors, as direct processing of stool donations does not seem necessary provided the fecal material is stored anaerobically.

In this study, we performed several culturing techniques combined with 16SrRNA sequencing to compare FMT preparation protocols with regard to bacterial viability. A strength of the study is that we compared current FMT protocols and assessed both short-term- and long-term-stored samples. Furthermore, we performed multiple culturing techniques in two distinct academic centers and reported similar results, minimizing potential batch effects and biases due to the methods used. We acknowledge a limitation of culturing, because it is well established that not all fecal organisms are readily culturable with conventional techniques [50,51]. Moreover, the type of media applied for culturing strongly determines which organisms can grow and, correspondingly, also determines the growth of other bacteria within trophic networks. Therefore, the diversity of the viable bacteria is probably underestimated. However, in our experiments with customized media for anaerobes, approximately 80% of bacterial genera that were detected in the uncultured frozen FMT material were also detected after plating, and most of the genera that were lost on plating were extremely low in abundance to start with. This indicates that the current results are representative for the majority of the microbial lineages present in these samples. Furthermore, our primary aim was to evaluate the influence of atmospheric oxygen on the bacterial viability of fecal samples, especially of anaerobes of interest (e.g., Clostridia belonging to Clusters IV and XIV), which we could successfully target. By comparing different processing conditions on subsamples from one donor at a time, we controlled for potential variance between fecal samples and were able to generalize our findings.

Currently, a clinical randomized controlled trial (RCT) to analyze the efficacy of anaerobically processed FMT on the induction of remission in active UC patients is ongoing (NCT05998213). Two previous RCTs that focused on anaerobic FMT for this indication yielded conflicting results [7,10], of which the negative trial did not yet publish information on the methods used. Future trials should put greater effort into analyzing the amount and types of viable bacterial donor taxa that are transplanted, and thus which metabolic capacity can be retained. This deeper knowledge is crucial for a better understanding of the mechanisms that contribute to FMT success. In addition to other factors such as antibiotic pretreatment, baseline microbiota signature, and recipients’ and donors’ diets, deeper knowledge of maintaining microbial viability is crucial for a better understanding of the mechanisms that contribute to FMT success.

In conclusion, our results show that processing fecal material in atmospheric oxygen adversely affects the number and composition of the viable bacteria, and therefore may limit clinical outcomes of FMT with regard to anti-inflammatory/immunomodulatory responses for the treatment of UC and other indications. Our results further suggest that, preferably, anaerobic processing of donor stool with short-term storage of frozen material should be applied for clinical use, in order to take advantage of the retention of obligate anaerobes and their metabolic products.

## Figures and Tables

**Figure 1 microorganisms-11-02238-f001:**
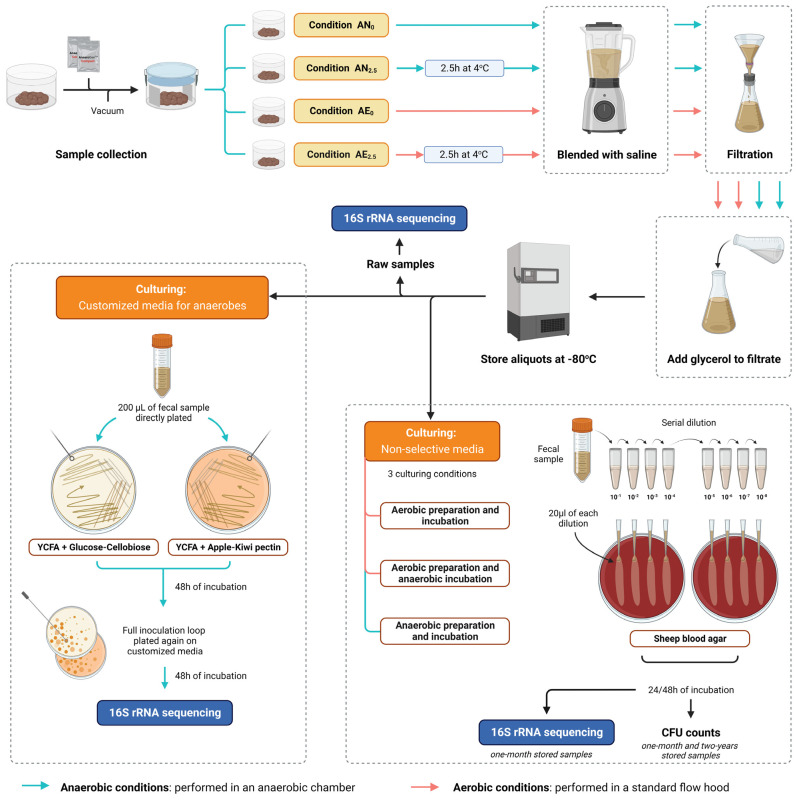
Schematic representation of the workflow used in this study.

**Figure 2 microorganisms-11-02238-f002:**
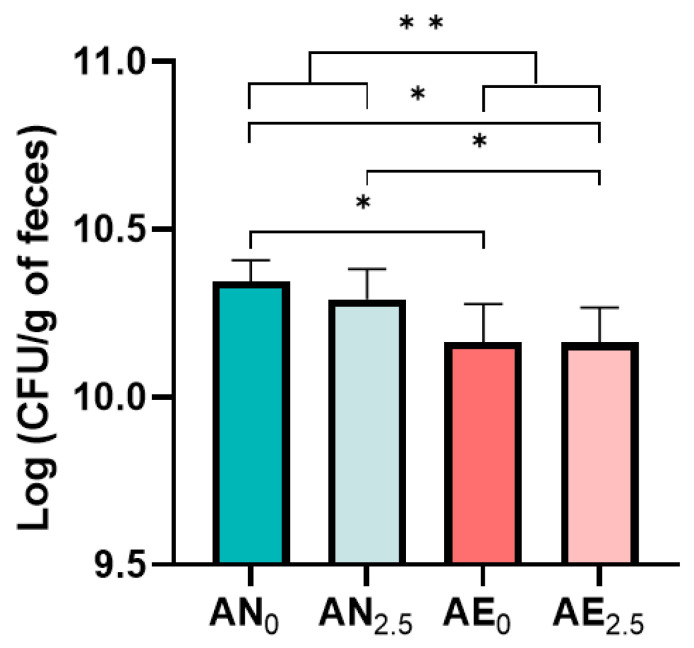
Median colony-forming-unit (CFU) counts per gram of feces of human stool samples (#1–#8) after strict anaerobic culturing on routine sheep blood agar plates, comparing four feces processing conditions for the production of frozen fecal microbiota suspensions. Statistical significance (*p*-value): <0.05 (*), <0.01 (**).

**Figure 3 microorganisms-11-02238-f003:**
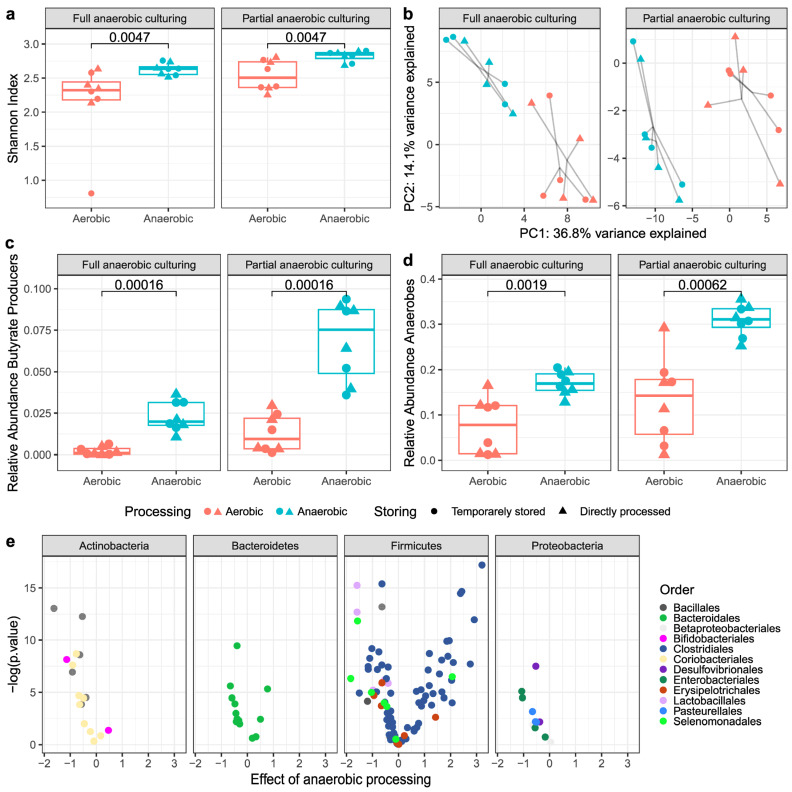
The influence of anaerobic and aerobic processing of human stool samples (#5–#8) on viable bacteria in fecal microbial transplant, as assessed by anaerobic culturing on routine sheep blood agar media and 16S rRNA sequencing. Two techniques of culturing were applied: partially anaerobic conditions, i.e., preparation in atmospheric oxygen and incubation under anaerobic conditions using AnaeroPack^®^ and full anaerobic conditions, i.e., preparation and incubation under anaerobic conditions. “Anaerobic” represents both anaerobic conditions during fecal suspension processing (AN_0_ and AN_2.5_), and “Aerobic” represents aerobic conditions (AE_0_ and AE_2.5_). (**a**) The α-diversity (Shannon index) of viable bacteria was significantly higher in the stool samples processed in anaerobic conditions compared with aerobic conditions, after both full and partial anaerobic culturing. (**b**) Multilevel principal component analysis (mPCA) displaying the beta-diversity (clr-transformed Euclidian distance) of viable bacterial communities. There is a clear division between samples processed in anaerobic conditions and samples processed in aerobic conditions for both culturing techniques. Temporary storage did not result in a clear composition shift of viable bacteria, as both anaerobically and aerobically stored samples cluster close to their respective directly processed samples. (**c**) Anaerobic processing resulted in a higher relative abundance of butyrate producers compared with aerobic processing. (**d**) Obligate anaerobes were more abundant in anaerobically processed fecal microbiota transplant compared with aerobically processed material with both culturing techniques. (**e**) Volcano plots showing bacterial orders within four major phyla associated with the fecal microbiota transplant processing method. Positive estimates on the *x*-axis, derived from the linear mixed models, associate with the anaerobically processed stool samples, whereas negative estimates associate with aerobically processed stool samples (both direct-processed and temporarily stored). Anaerobic processing was associated with higher abundances of Clostridiales (phylum Firmicutes), whereas aerobic processing was associated with higher abundances of Coriobacteriales (phylum Actinobacteria) and Enterobacteriales (phylum Proteobacteria). *Abbreviations: GC: GC medium* (*YCFA medium with glucose—cellobiose*), *P: P medium* (*YCFA medium with pectines*).

**Figure 4 microorganisms-11-02238-f004:**
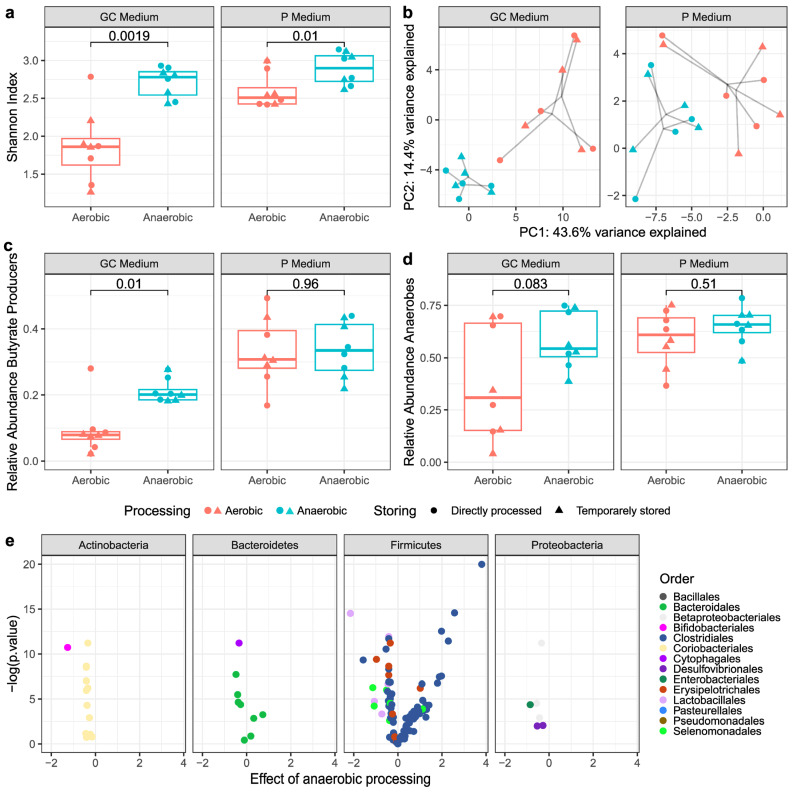
The influence of anaerobic and aerobic processing of human stool samples (#1–#4) on viable bacteria in fecal microbial transplant, as assessed by anaerobic culturing on two types of customized media for anaerobes and 16S rRNA sequencing. “Anaerobic” represents both anaerobic conditions during fecal suspension processing (AN_0_ and AN_2.5_), and “Aerobic” represents aerobic conditions (AE_0_ and AE_2.5_). (**a**) The α-diversity (Shannon index) of viable bacteria was significantly higher in the stool samples processed in anaerobic conditions compared with aerobic conditions, after culturing both on GC medium and P medium. (**b**) Multilevel principal component analysis (mPCA) displaying the beta-diversity (clr-transformed Euclidian distance) of viable bacterial communities. There is a clear division between samples processed in anaerobic conditions and samples processed in aerobic conditions. Temporary storage did not result in a clear composition shift of viable bacteria, as both anaerobically and aerobically stored samples cluster close to their respective directly processed samples. (**c**) Anaerobic processing resulted in a higher relative abundance of butyrate producers compared with aerobic processing after culturing on GC medium. Anaerobic culturing on P medium resulted in an overall higher abundance of butyrate producers, with no significant difference between anaerobically and aerobically processed samples. (**d**) A trend toward a higher abundance of viable obligate anaerobes after anaerobic processing was observed for both media. (**e**) Volcano plots showing bacterial orders within four major phyla associated with the fecal microbiota transplant processing method. Positive estimates on the *x*-axis, derived from the linear mixed models, associate with the anaerobically processed stool samples, whereas negative estimates associate with aerobically processed stool samples (both direct and temporarily stored). Anaerobic processing associated with higher abundances of Clostridiales (phylum Firmicutes), whereas aerobic processing associated with higher abundances of Coriobacteriales (phylum Actinobacteria), Betaproteobacteriales and Desulfovibrionales (phylum Proteobacteria). *Abbreviations: GC: GC medium* (*YCFA medium with glucose—cellobiose*), *P: P medium* (*YCFA medium with pectines*).

## Data Availability

Data can be retrieved here: https://www.ebi.ac.uk/ena/browser/view/PRJEB63205. Accessed on 15 June 2023.

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
