# Peer review of "Anaerobic Feces Processing for Fecal Microbiota Transplantation Improves Viability of Obligate Anaerobes"

_microorganisms, 2023, doi:10.3390/microorganisms11092238_

Round 1
Reviewer 1 Report
Interesting study to assess bacterial viability of FMT products by aerobic and anaerobic manufacturing processes, and combined both culturing and sequencing techniques, which is a strength of the study.
However, I think several improvements can be made.
1. Introduction: since FMT is predominant done for recurrent CDI (rCDI), the authors should at least mention this indication here instead in discussion, and how the success of FMT is not affected by aerobic processing. This also argues for the fact that viable bacteria may not be the most crucial element to explain the clinical efficacy in rCDI; however, it may be different in other indications.
2. Median of 10.31 log CFU/g feces vs 10.16 log CFU/g feces may be statistically significant, but does this translate into clinically meaningful difference when it comes to bacterial engraftment and clinical efficacy?
3. After one month storage, there was a statistically significant difference between AN and AE, whereas there is no difference between AN and AE FMT products. However, the actual numbers of CFU seemed to be quite similar for all AN and AE products, which is interesting, and suggests that 2 year storage does not affect viability whether manufactured anaerobically or aerobically, still in the range of 1010 if we just look at CFU. The difference in composition between the AN and AE seems to diminish following long term storage. I think this is a point worth emphasizing.
4. Conclusion: several RCTs using FMT to treat UC have reported efficacy in the range of 30-50%, whether FMT is manufactured aerobically or anaerobically. Therefore, bacterial viability of the donor may not fully explain the efficacy. There are other ecological principles which may govern engraftment, such as competition, exclusion etc. Furthermore, diet patterns of the donors or recipients have not been accounted for in majority of these studies, which makes it difficult to make conclusions. I think it should be emphasized again that viability is not the only factor to consider.
5. minor issue: typo: line 173 on page 5- duplo should be corrected to duplicates
Author Response
Interesting study to assess bacterial viability of FMT products by aerobic and anaerobic manufacturing processes, and combined both culturing and sequencing techniques, which is a strength of the study. However, I think several improvements can be made.
Point 1. Introduction: since FMT is predominant done for recurrent CDI (rCDI), the authors should at least mention this indication here instead in discussion, and how the success of FMT is not affected by aerobic processing. This also argues for the fact that viable bacteria may not be the most crucial element to explain the clinical efficacy in rCDI; however, it may be different in other indications.
Response 1. We now added the primary use of FMT for rCDI in the first paragraph of the introduction even as the fact that the clinical efficacy for this use does not seem to depend on stool processing protocols and donor microbiota composition.
Point 2. Median of 10.31 log CFU/g feces vs 10.16 log CFU/g feces may be statistically significant, but does this translate into clinically meaningful difference when it comes to bacterial engraftment and clinical efficacy?
Response 2. We agree with the reviewer that a statistically significant difference in quantitative viable bacteria does not infer clinical relevance. Currently we are conducting a randomized controlled trial that uses directly anaerobic processed FMT (TURN2 trial) to further answer this question. We address this in the discussion (Revised manuscript: line 450-451).
Point 3. After one month storage, there was a statistically significant difference between AN and AE, whereas there is no difference between AN and AE FMT products. However, the actual numbers of CFU seemed to be quite similar for all AN and AE products, which is interesting, and suggests that 2 year storage does not affect viability whether manufactured anaerobically or aerobically, still in the range of 1010 if we just look at CFU. The difference in composition between the AN and AE seems to diminish following long term storage. I think this is a point worth emphasizing.
Response 3. We thank the reviewer for this valid point. With linear mixed model analysis we found that indeed the atmosphere (anaerobic vs. aerobic) had an interaction effect with storage time; showing that the effect of atmosphere was larger for one-month stored samples and smaller for two-years stored samples. This is stated in the results (Revised manuscript: line 266-270). Similarly, we did not find significant differences between the processing conditions when we only analyzed the two-years stored samples (Revised manuscript: line 271-275). In the discussion we discuss this point in lines 397-402. To further emphasize this point we added a potential implication of this finding (Revised manuscript: lines 402 – 404).
Point 4. Conclusion: several RCTs using FMT to treat UC have reported efficacy in the range of 30-50%, whether FMT is manufactured aerobically or anaerobically. Therefore, bacterial viability of the donor may not fully explain the efficacy. There are other ecological principles which may govern engraftment, such as competition, exclusion etc. Furthermore, diet patterns of the donors or recipients have not been accounted for in majority of these studies, which makes it difficult to make conclusions. I think it should be emphasized again that viability is not the only factor to consider.
Response 4. We agree with the reviewer that viability is only one of the potential factors influencing FMT efficacy. We added this in the discussion (Lines 459-462) to make a nuance to the importance of bacterial viability.
Point 5. minor issue: typo: line 173 on page 5- duplo should be corrected to duplicates.
Response 5. This point has been adjusted in the manuscript.

Reviewer 2 Report
The paper by Mèlanie V. Bénard et al. entitled "Anaerobic feces processing for fecal microbiota transplantation improves the viability of obligate anaerobes" is an interesting study on the impact of anaerobic preparations on the viability of obligate anaerobes in the stool preparations for fecal microbiota transplantation (FMT). This paper is well-written, and well-designed study protocols present meaningful data. It will be of interest to readers of this Journal. However, I have the following concerns.
Major comments:
1. The peer review file contained only Supplemental Figures 1a, 1b, 1c, and Supplemental Tables 4 and 5. Supplemental Tables 1, 2, and 3 were missing. The peer review was conducted without the information provided in these tables at this time.
2. The study is very well designed and shows interesting results. However, the study is rather complex, and the study design is somewhat difficult to understand.
3. In the section "Microbial culturomics using routine non-selective media," the authors state that #1-#4 and #5-#8 were cultured on Columbia sheep blood agar plates (lines 141-142). However, Figure 3 only shows the results of one-month storage samples #5-#8. Were samples #1-#4 not cultured on non-selective media? If this is the case, please clarify in this section that cultures by non-selective media were performed on 1-month storage samples #5-#8.
4. In addition, for better reader understanding, please consider adding to Figure 1 that #1-#4 are two-year storage samples and #5-#8 are one-month storage samples, and that #1-#4 were cultured on non-selective media and #5-#8 were cultured on customized media.
5. Figures 3A-3D compare the "aerobic" and "anaerobic" by dividing them into "full anaerobic" and "partial anaerobic." Is "full anaerobic" referring to "anaerobic preparation and incubation (i.e., anaerobic chamber Concept 500 and AnaeroPack®)"? Similarly, "partial anaerobic" refers to "aerobic preparation and anaerobic incubation (i.e., preparation in atmospheric oxygen and anaerobic incubation under anaerobic conditions using AnaeroPack®)"? If this understanding is correct, please make sure that the "full anaerobic" and "partial anaerobic" results in Figures 3A-3D are not presented in reverse." I thought "full anaerobic" might have a higher Shannon index and relative abundance of butyrate producer than "partial anaerobic." If this figure is correct, please mention why this reversal has occurred.
6. Why is the data on "aerobic conditions (i.e., flow hood and incubator)" not shown?
Minor comments
1. Please change "AN" and "AE" in Figure 2 to "AN0" and "AE0", respectively, as described in the text.
Author Response
Major comments:
Point 1. The peer review file contained only Supplemental Figures 1a, 1b, 1c, and Supplemental Tables 4 and 5. Supplemental Tables 1, 2, and 3 were missing. The peer review was conducted without the information provided in these tables at this time.
Response 1. We apologize for the inconvenience and thank the reviewer for the notice. We have send the adjusted zip-file with all supplementary figures and tables along with this revision.
Point 2. The study is very well designed and shows interesting results. However, the study is rather complex, and the study design is somewhat difficult to understand.
Response 2. We acknowledge that the study design is complex since we conducted two separate experiments in two University Medical Centers with distinct methods for culturing, DNA isolation and 16S rRNA sequencing. Therefore we made an extensive Figure in Biorender to guide the reader through the experiments. To increase readability, we now start the method section with a reference to this figure. Despite the complexity as a result of multiple experiments, we believe that this workflow contributes to the generalizability of our results. In addition, we now emphasized the difference between the aerobic and anaerobic processing and the aerobic and anerobic culturing more clearly to avoid confusion.
Point 3. In the section "Microbial culturomics using routine non-selective media," the authors state that #1-#4 and #5-#8 were cultured on Columbia sheep blood agar plates (lines 141-142). However, Figure 3 only shows the results of one-month storage samples #5-#8. Were samples #1-#4 not cultured on non-selective media? If this is the case, please clarify in this section that cultures by non-selective media were performed on 1-month storage samples #5-#8.
Response 3. We thank the reviewer for this comment. Quantification of bacteria (CFU counts) was performed on eight samples (short-term and long-term stored samples) after culturing on routine media. We changed the initial subheading to ‘’Quantification of live bacteria with CFU counts in short-term and long-term stored samples using routine non-selective media’’ to clarify and make it compatible with the title of the corresponding result section 3.1. 16S rRNA sequencing was performed after culturing on specialized media (for sample #1-4 after short storage) and after culturing on routine media (only for sample #5-8 after short storage). We point the reviewer to a latter section in the methods where this is stated (line 178) in the section ‘’DNA extraction, PCR, and 16S rRNA sequencing’’. The storage times for CFU counts are also stated in Supplementary Table 1.
Point 4. In addition, for better reader understanding, please consider adding to Figure 1 that #1-#4 are two-year storage samples and #5-#8 are one-month storage samples, and that #1-#4 were cultured on non-selective media and #5-#8 were cultured on customized media.
Response 4. Thank you for your recommendation. Only for the CFU counts we used short-term and long-term stored samples. The sequencing was done in both experiments with short term stored samples. To make this more clear we added in Figure 1 ‘’one-month and two-years stored samples’’ underneath the CFU counts and ‘’one-month stored samples’’ underneath 16S rRNA sequencing.
Point 5. Figures 3A-3D compare the "aerobic" and "anaerobic" by dividing them into "full anaerobic" and "partial anaerobic." Is "full anaerobic" referring to "anaerobic preparation and incubation (i.e., anaerobic chamber Concept 500 and AnaeroPack®)"? Similarly, "partial anaerobic" refers to "aerobic preparation and anaerobic incubation (i.e., preparation in atmospheric oxygen and anaerobic incubation under anaerobic conditions using AnaeroPack®)"? If this understanding is correct, please make sure that the "full anaerobic" and "partial anaerobic" results in Figures 3A-3D are not presented in reverse." I thought "full anaerobic" might have a higher Shannon index and relative abundance of butyrate producer than "partial anaerobic." If this figure is correct, please mention why this reversal has occurred.
Response 5. Thank you for your question. The fully and partially anaerobic culture conditions refer to the culturing conditions as you mention correctly, the ‘anaerobic’ and ‘aerobic’ terms to the oxygen condition during fecal microbiota suspension preparation. We did not show a direct comparison between the fully and partially anaerobic culture conditions since we’re interested in any differences resulting in the preparation process. The difference between fully and partially anaerobic culturing is a technical difference. The results in 3A-3D are not presented in reverse.
We realized that the headings and definitions on the x-axes are unclear. Therefore we changed the headings to ‘’full anaerobic culturing’’ and ‘’partial anaerobic culturing’’. Lastly, the definitions of the two culturing techniques are now clarified in the legend of Figure 3, and the 2 processing conditions are explained in the legend of both Figure 3 and Figure 4.
Point 6. Why is the data on "aerobic conditions (i.e., flow hood and incubator)" not shown?
Response 6. We chose not to show the data on the aerobic conditions in the manuscript since these data are not essential for our main research question; that is if anaerobic preparation of fecal microbiota suspensions do preserve anaerobic bacteria better compared to aerobic preparation methods. After fully aerobic preparation we expect way less CFU counts since the majority of gut microbiota consist of anaerobes (biological difference). In line with expectations, we found low CFU counts in the fully aerobic condition. Because we acknowledge our current results are already quite complex to understand, we choose to only show essential data for our research question in the main text. The exact CFU counts from the aerobic culturing are however presented in the supplementary files (supplementary table 2).
We now added visual results of the different culturing techniques to the supplementary files (Figure 2). We refer to this figure in lines 285-287.
Mean colony forming unit (CFU) counts per gram of feces of stool sample #1 - #8, comparing three culturing conditions and four feces processing conditions. Aerobic culturing resulted in lower CFU counts for all processing conditions compared to (full and partial) anaerobic culturing.
Significance (p-value): p< 0.0001 (****).
Minor comments
Point 7. Please change "AN" and "AE" in Figure 2 to "AN0" and "AE0", respectively, as described in the text.
Response 7. This point has been adjusted in the manuscript.

Reviewer 3 Report
General Comments
The publication "Anaerobic feces processing for fecal microbiota transplantation improves viability of obligate anaerobes" presents original results aimed at highlighting the survival of bacterial populations present in the stool during the process of treatment and preservation of these. The results and the proposed methodologies are very interesting and really advance knowledge in this important field for the future. The article is well written, clear and complete. It deserves some additional explanations concerning the conditioning and storage of samples at -80°C, on the models for studying viability and on the legend of two figures (see specific remarks below)
Specific comments
Line 90: Replace "recurrent Clostridioides difficile (rCDI)" with "recurrent Clostridioides difficile infection (rCDI)"
Lines 132-137: Specify the protocol followed, in particular whether the manipulations were carried out under aerobiosis or anaerobiosis, whether the saline solution was previously degassed, the type of packaging and the volumes of the aliquots stored at -80°C and the presence or no oxygen in this one.
Figure 1: Specify whether the blender, filtration and glycerol steps are carried out in contact with air and the type of packaging of the stored aliquots. Is the inoculation of the 20 microlitrers on "sheep blood agar" really carried out as shown in the figure.
Lines 210-211: specify the type of model used.
Line 310: Replace "Clostridialis" with "Clostridiales"
Figure 3: Clarify in the legend the meaning of “Full anaerobic”, “Partial anaerobic”, “Aerobic” and “Anaerobic” as it is not clear.
Figures 3 and 4: Do not put only the small circles for the color legend but also the small triangles. The letters (A to E) are in upper case in the figure and in lower case in the legend. The legend of part (e) is not clear and should be reworded: aero-anaerobic bacteria like proteobacteria seem favored by the anaerobic protocol, for example. Moreover, the colors used in the volcano-plots is not very discriminating.
Lines 384-386: discuss type of aliquot preparation and packaging process.
Lines 415-424: add an argument concerning the fact that the subcultures favor the flora most adapted to the culture conditions used and therefore that the real diversity is probably higher than that found by the method proposed to assess viability.
Author Response
The publication "Anaerobic feces processing for fecal microbiota transplantation improves viability of obligate anaerobes" presents original results aimed at highlighting the survival of bacterial populations present in the stool during the process of treatment and preservation of these. The results and the proposed methodologies are very interesting and really advance knowledge in this important field for the future. The article is well written, clear and complete. It deserves some additional explanations concerning the conditioning and storage of samples at -80°C, on the models for studying viability and on the legend of two figures (see specific remarks below)
Specific comments
Point 1. Line 90: Replace "recurrent Clostridioides difficile (rCDI)" with "recurrent Clostridioides difficile infection (rCDI)"
Response 1. This point has been adjusted in the manuscript.
Point 2. Lines 132-137: Specify the protocol followed, in particular whether the manipulations were carried out under aerobiosis or anaerobiosis, whether the saline solution was previously degassed, the type of packaging and the volumes of the aliquots stored at -80°C and the presence or no oxygen in this one.
Response 2. Samples were stored in 60cc Nutrifit syringes with cap and wrapped with aluminium foil. This storage material and used saline and glycerol were placed inside the anaerobic chamber at least 48 hours before use. These specifics were added to the methods section.
Point 3. Figure 1: Specify whether the blender, filtration and glycerol steps are carried out in contact with air and the type of packaging of the stored aliquots. Is the inoculation of the 20 microlitrers on "sheep blood agar" really carried out as shown in the figure.
Response 3. With regards to the blending, filtration and glycerol step we kindly refer the reviewer to the blue and red arrows representing respectively anaerobic preparation (for condition AN and AN2.5) and aerobic preparation (for condition AE0 and AE0). We chose not to add the used packaging to Figure 1 for legibility, but added this in the text in lines 139-140. The inoculation on the sheep blood agar was carried out as represented in Figure 1 (with 4 vertical smears).
Point 4. Lines 210-211: specify the type of model used.
Response 4. The details of the model are added to the methods and legend.
Point 5. Line 310: Replace "Clostridialis" with "Clostridiales"
Response 5. This typo has been adjusted in the manuscript.
Point 6. Figure 3: Clarify in the legend the meaning of “Full anaerobic”, “Partial anaerobic”, “Aerobic” and “Anaerobic” as it is not clear.
Response 6. We agree with the reviewer that the headings and definitions on the x-axes are unclear. Therefore we changed the headings to ‘’full anaerobic culturing’’ and ‘’partial anaerobic culturing’’. Lastly, the definitions of the two culturing techniques are now clarified in the legend of Figure 3, and the 2 processing conditions are explained in the legend of both Figure 3 and Figure 4.
Point 7. Figures 3 and 4: Do not put only the small circles for the color legend but also the small triangles. The letters (A to E) are in upper case in the figure and in lower case in the legend. The legend of part (e) is not clear and should be reworded: aero-anaerobic bacteria like proteobacteria seem favored by the anaerobic protocol, for example. Moreover, the colors used in the volcano-plots is not very discriminating.
Response 7. Thank you for your suggestions. We changed the color scheme of the volcano plot, and altered the legends.
Point 8. Lines 384-386: discuss type of aliquot preparation and packaging process.
Response 8. We choose to add details on aliquot preparation and packaging in the methods, lines 139-141.
Point 9. Lines 415-424: add an argument concerning the fact that the subcultures favor the flora most adapted to the culture conditions used and therefore that the real diversity is probably higher than that found by the method proposed to assess viability.
Response 9. We added this to the discussion section.
